# A Neural Representation of Sketch Drawings

**David Ha**
Google Brain
hadavid@google.com

**Douglas Eck**
Google Brain
deck@google.com

## Abstract

We present `sketch-rnn`, a recurrent neural network (RNN) able to construct stroke-based drawings of common objects. The model is trained on a dataset of human-drawn images representing many different classes. We outline a framework for conditional and unconditional sketch generation, and describe new robust training methods for generating coherent sketch drawings in a vector format.

## 1 Introduction

Recently, there have been major advancements in generative modelling of images using neural networks as a generative tool. Generative Adversarial Networks (GANs) (Goodfellow, 2016), Variational Inference (VI) (Kingma & Welling, 2013), and Autoregressive (AR) (Reed et al., 2017) models have become popular tools in this fast growing area. Most of the work thus far has been targeted towards modelling low resolution, pixel images. Humans, however, do not understand the world as a grid of pixels, but rather develop abstract concepts to represent what we see. From a young age, we develop the ability to communicate what we see by drawing on paper with a pencil or crayon. In this way we learn to express a sequential, vector representation of an image as a short sequence of strokes. In this paper we investigate an alternative to traditional pixel image modelling approaches, and propose a generative model for vector images.

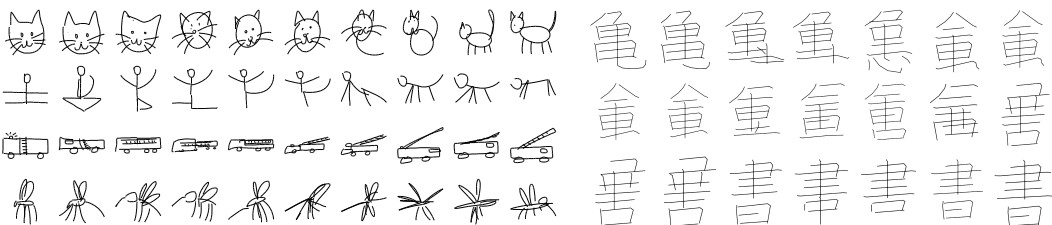

Figure 1: Latent space interpolation of various vector images produced by our model (left). Interpolation of two different Kanji characters (亀→書) as sequence of strokes (right).

Our goal is to train machines to draw and generalize abstract concepts in a manner similar to humans. In this work, as a first step towards this goal, we train our model on a dataset of hand-drawn sketches, each represented as a sequence of motor actions controlling a pen: which direction to move, when to lift the pen up, and when to stop drawing. In doing so, we created a model that potentially has many applications, from assisting the creative process of an artist, to helping teach students how to draw.

This paper makes the following contributions: We outline a framework for both unconditional and conditional generation of vector images composed of a sequence of lines. Our recurrent neural network-based generative model is capable of producing sketches of common objects in a vector format. We develop a training procedure unique to vector images to make the training more robust. In the conditional generation model, we explore the latent space developed by the model to represent a vector image. We also discuss creative applications of our methodology. We make available a dataset of 50 million hand drawn vector images to encourage further development of generative modelling for vector images, and also release an implementation of our model as an open source project.[1]

---

[1] The code and dataset is available at `https://magenta.tensorflow.org/sketch_rnn`.

## 2 RELATED WORK

There is a long history of work related to algorithms that mimic painters. One such work is Portrait Drawing by Paul the Robot (Tresset & Fol Leymarie, 2013; Xie et al., 2012), where an underlying algorithm controlling a mechanical robot arm sketches lines on a canvas with a programmable artistic style to mimic a given digitized portrait of a person. Reinforcement Learning based-approaches (Xie et al., 2012) have been developed to discover a set of paint brush strokes that can best represent a given input photograph. These prior works generally attempt to mimic digitized photographs, rather than develop generative models of vector images.

Neural Network-based approaches have been developed for generative models of images, although the majority of neural network-related research on image generation deal with pixel images (Goodfellow, 2016; Isola et al., 2016; Kaae Sønderby et al., 2016; Kingma et al., 2016; Reed et al., 2017; White, 2016). There has been relatively little work done on vector image generation using neural networks. An earlier work (Simhon & Dudek, 2004) makes use of Hidden Markov Models to synthesize lines and curves of a human sketch. More recent work (Graves, 2013) on handwriting generation with Recurrent Neural Networks laid the groundwork for utilizing Mixture Density Networks (Bishop, 1994) to generate continuous data points. Recent works of this approach attempted to generate vectorized Kanji characters (Ha, 2015; Zhang et al., 2016) by modelling Chinese characters as a sequence of pen stroke actions.

The approach outlined in this work allows one to explore the latent space representation of vector images. For instance, we can use our model to interpolate between two Kanji characters in Figure 1 by first encoding the characters, represented as a sequence of strokes, into a latent space of embedding vectors. Previous work (Bowman et al., 2015) outlined a methodology to combine Sequence-to-Sequence models with a Variational Autoencoder to model natural English sentences in latent vector space. A related work (Lake et al., 2015), utilizes probabilistic program induction, rather than neural networks, to perform one-shot modelling of the Omniglot dataset containing images of symbols.

One of the factors limiting research development in the space of generative vector drawings is the lack of publicly available datasets. Previously, the Sketch dataset (Eitz et al., 2012), consisting of 20K vector sketches, was used to explore feature extraction techniques. A subsequent work, the Sketchy dataset (Sangkloy et al., 2016), provided 70K vector sketches along with corresponding pixel images for various classes. This allowed for a larger-scale exploration of human sketches. ShadowDraw (Lee et al., 2011) is an interactive system that predicts what a finished drawing looks like based on a set of incomplete brush strokes from the user while the sketch is being drawn. ShadowDraw used a dataset of 30K raster images combined with extracted vectorized features. In this work, we use a much larger dataset of 50 million vector sketches that is made publicly available.

## 3 METHODOLOGY

### 3.1 DATASET

We constructed `QuickDraw`, a dataset of 50 million vector drawings obtained from *Quick, Draw!* (Jongejan et al., 2016), an online game where the players are asked to draw objects belonging to a particular object class in less than 20 seconds. `QuickDraw` consists of hundreds of classes of common objects. Each class of `QuickDraw` is a dataset of 70K training samples, in addition to 2.5K validation and 2.5K test samples.

We use a data format that represents a sketch as a set of pen stroke actions. This representation is an extension of the format used in (Graves, 2013). Our format extends the binary pen stroke event into a multi-state event. In this data format, the initial coordinate of the drawing is located at the origin.

A sketch is a list of points, and each point is a vector consisting of 5 elements: $(\Delta x, \Delta y, p_1, p_2, p_3)$. The first two elements are the offset distance in the x and y directions of the pen from the previous point. The last 3 elements represents a binary one-hot vector of 3 possible states. The first pen state, $p_1$, indicates that the pen is currently touching the paper, and that a line will be drawn connecting the next point with the current point. The second pen state, $p_2$, indicates that the pen will be lifted from the paper after the current point, and that no line will be drawn next. The final pen state, $p_3$, indicates that the drawing has ended, and subsequent points, including the current point, will not be rendered.

## 3.2 SKETCH-RNN

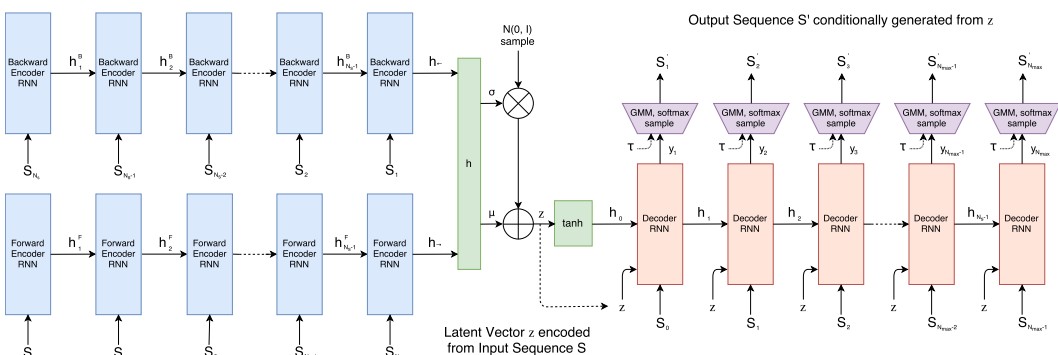

Figure 2: Schematic diagram of `sketch-rnn`.

Our model is a Sequence-to-Sequence Variational Autoencoder (VAE), similar to the architecture described in (Bowman et al., 2015; Kingma & Welling, 2013). Our encoder is a bidirectional RNN (Schuster et al., 1997) that takes in a sketch as an input, and outputs a latent vector of size $N_z$. Specifically, we feed the sketch sequence, $S$, and also the same sketch sequence in reverse order, $S_{\text{reverse}}$, into the two encoding RNNs of the bidirectional RNN, to obtain two final hidden states:

$$h_\rightarrow = \texttt{encode}_\rightarrow(S), \ h_\leftarrow = \texttt{encode}_\leftarrow(S_{\text{reverse}}), \ h = [\, h_\rightarrow \, ; \, h_\leftarrow \,]. \tag{1}$$

We take this final concatenated hidden state, $h$, and project it into two vectors $\mu$ and $\hat{\sigma}$, each of size $N_z$, using a fully connected layer. We convert $\hat{\sigma}$ into a non-negative standard deviation parameter $\sigma$ using an exponential operation. We use $\mu$ and $\sigma$, along with $\mathcal{N}(0, I)$, a vector of IID Gaussian variables of size $N_z$, to construct a random vector, $z \in \mathbb{R}^{N_z}$, as in the approach for a VAE:

$$\mu = W_\mu h + b_\mu, \ \hat{\sigma} = W_\sigma h + b_\sigma, \ \sigma = \exp\left(\frac{\hat{\sigma}}{2}\right), \ z = \mu + \sigma \odot \mathcal{N}(0, I). \tag{2}$$

Under this encoding scheme, the latent vector $z$ is not a deterministic output for a given input sketch, but a random vector conditioned on the input sketch.

Our decoder is an autoregressive RNN that samples output sketches conditional on a given latent vector $z$. The initial hidden states $h_0$, and optional cell states $c_0$ (if applicable) of the decoder RNN is the output of a single layer network: $[\, h_0 \, ; \, c_0 \,] = \tanh(W_z z + b_z)$

At each step $i$ of the decoder RNN, we feed the previous point, $S_{i-1}$ and the latent vector $z$ in as a concatenated input $x_i$, where $S_0$ is defined as $(0, 0, 1, 0, 0)$. The output at each time step are the parameters for a probability distribution of the next data point $S_i$. In Equation 3, we model $(\Delta x, \Delta y)$ as a Gaussian mixture model (GMM) with $M$ normal distributions as in (Bishop, 1994; Graves, 2013), and $(q_1, q_2, q_3)$ as a categorical distribution to model the ground truth data $(p_1, p_2, p_3)$, where $(q_1 + q_2 + q_3 = 1)$ as done in (Ha, 2015) and (Zhang et al., 2016). Unlike (Graves, 2013), our generated sequence is conditioned from a latent code $z$ sampled from our encoder, which is trained end-to-end alongside the decoder.

$$p(\Delta x, \Delta y) = \sum_{j=1}^{M} \Pi_j \, \mathcal{N}(\Delta x, \Delta y \mid \mu_{x,j}, \mu_{y,j}, \sigma_{x,j}, \sigma_{y,j}, \rho_{xy,j}), \text{ where } \sum_{j=1}^{M} \Pi_j = 1 \tag{3}$$

$\mathcal{N}(x, y | \mu_x, \mu_y, \sigma_x, \sigma_y, \rho_{xy})$ is the probability distribution function for a bivariate normal distribution. Each of the $M$ bivariate normal distributions consist of five parameters: $(\mu_x, \mu_y, \sigma_x, \sigma_y, \rho_{xy})$, where $\mu_x$ and $\mu_y$ are the means, $\sigma_x$ and $\sigma_y$ are the standard deviations, and $\rho_{xy}$ is the correlation parameter of each bivariate normal distribution. An additional vector $\Pi$ of length $M$, also a categorical distribution, are the mixture weights of the Gaussian mixture model. Hence the size of the output vector $y$ is $5M + M + 3$, which includes the 3 logits needed to generate $(q_1, q_2, q_3)$.

The next hidden state of the RNN, generated with its forward operation, projects into the output vector $y_i$ using a fully-connected layer:

$$x_i = [\, S_{i-1} \, ; \, z \,], \ [\, h_i \, ; \, c_i \,] = \texttt{forward}(x_i, [\, h_{i-1} \, ; \, c_{i-1} \,]), \ y_i = W_y h_i + b_y, \ y_i \in \mathbb{R}^{6M+3}. \tag{4}$$

The vector $y_i$ is broken down into the parameters of the probability distribution of the next data point:

$$[\,(\hat{\Pi}\ \mu_x\ \mu_y\ \hat{\sigma}_x\ \hat{\sigma}_y\ \hat{\rho}_{xy})_1\ (\hat{\Pi}\ \mu_x\ \mu_y\ \hat{\sigma}_x\ \hat{\sigma}_y\ \hat{\rho}_{xy})_2\ ...\ (\hat{\Pi}\ \mu_x\ \mu_y\ \hat{\sigma}_x\ \hat{\sigma}_y\ \hat{\rho}_{xy})_M\ (\hat{q}_1\ \hat{q}_2\ \hat{q}_3)\,] = y_i. \quad (5)$$

As in (Graves, 2013), we apply exp and tanh operations to ensure the standard deviation values are non-negative, and that the correlation value is between -1 and 1:

$$\sigma_x = \exp(\hat{\sigma}_x),\ \sigma_y = \exp(\hat{\sigma}_y),\ \rho_{xy} = \tanh(\hat{\rho}_{xy}). \quad (6)$$

The probabilities for the categorical distributions are calculated using the outputs as logit values:

$$q_k = \frac{\exp(\hat{q}_k)}{\sum_{j=1}^{3} \exp(\hat{q}_j)}, k \in \{1,\ 2,\ 3\},\ \Pi_k = \frac{\exp(\hat{\Pi}_k)}{\sum_{j=1}^{M} \exp(\hat{\Pi}_j)}, k \in \{1,\ ...,\ M\}. \quad (7)$$

A key challenge is to train our model to know when to stop drawing. Because the probabilities of the three pen stroke events are highly unbalanced, the model becomes more difficult to train. The probability of a $p_1$ event is much higher than $p_2$, and the $p_3$ event will only happen once per drawing. The approach developed in (Ha, 2015) and later followed by (Zhang et al., 2016) was to use different weightings for each pen event when calculating the losses, such as a hand-tuned weighting of $(1, 10, 100)$. We find this inelegant approach to be inadequate for our dataset of diverse images.

We develop a simpler, more robust approach that works well for a broad class of sketch drawing data. In our approach, all sequences are generated to a length of $N_{\max}$ where $N_{\max}$ is the length of the longest sketch in our training dataset. In principle $N_{\max}$ can be considered a hyper parameter. As the length of $S$ is usually shorter than $N_{\max}$, we set $S_i$ to be $(0, 0, 0, 0, 1)$ for $i > N_s$. We discuss the training in detail in the next section.

After training, we can sample sketches from our model. During the sampling process, we generate the parameters for both GMM and categorical distributions at each time step, and sample an outcome $S'_i$ for that time step. Unlike the training process, we feed the sampled outcome $S'_i$ as input for the next time step. We continue to sample until $p_3 = 1$, or when we have reached $i = N_{\max}$. Like the encoder, the sampled output is not deterministic, but a random sequence, conditioned on the input latent vector $z$. We can control the level of randomness we would like our samples to have during the sampling process by introducing a temperature parameter $\tau$:

$$\hat{q}_k \rightarrow \frac{\hat{q}_k}{\tau},\ \hat{\Pi}_k \rightarrow \frac{\hat{\Pi}_k}{\tau},\ \sigma_x^2 \rightarrow \sigma_x^2 \tau,\ \sigma_y^2 \rightarrow \sigma_y^2 \tau. \quad (8)$$

We can scale the softmax parameters of the categorial distribution and also the $\sigma$ parameters of the bivariate normal distribution by a temperature parameter $\tau$, to control the level of randomness in our samples. $\tau$ is typically set between 0 and 1. In the limiting case as $\tau \rightarrow 0$, our model becomes deterministic and samples will consist of the most likely point in the probability density function. Figure 3 illustrates of effect of sampling sketches with various temperature parameters.

## 3.3 UNCONDITIONAL GENERATION

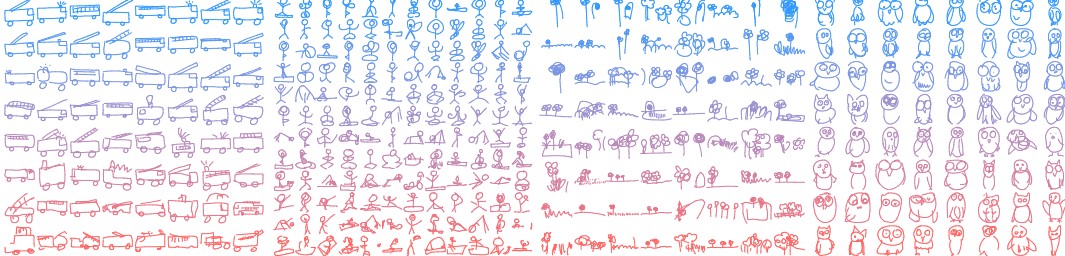

Figure 3: Unconditional generation of firetrucks, yoga poses, gardens and owls with varying $\tau$.

As a special case, we can also train our model to generate sketches unconditionally, where we only train the decoder RNN module, without any input or latent vectors. By removing the encoder, the decoder RNN as a standalone model is an autoregressive model without latent variables. In this use case, the initial hidden states and cell states of the decoder RNN are initialized to zero. The inputs $x_i$ of the decoder RNN at each time step is only $S_{i-1}$ or $S'_{i-1}$, as we do not need to concatenate a latent vector $z$. In Figure 3, we sample various sketch images generated unconditionally by varying the temperature parameter from $\tau = 0.2$ at the top in blue, to $\tau = 0.9$ at the bottom in red.

### 3.4 TRAINING

Our training procedure follows the approach of the Variational Autoencoder (Kingma & Welling, 2013), where the loss function is the sum of two terms: the Reconstruction Loss, $L_R$, and the Kullback-Leibler Divergence Loss, $L_{KL}$. We train our model to optimize this two-part loss function. The Reconstruction loss term, described in Equation 9, maximizes the log-likehood of the generated probability distribution to explain the training data $S$. We can calculate this reconstruction loss, $L_R$, using the generated parameters of the pdf and the training data $S$. $L_R$ is composed of the sum of the log loss of the offset terms $(\Delta x, \Delta y)$, $L_s$, and the log loss of the pen state terms $(p_1, p_2, p_3)$, $L_p$:

$$L_s = -\frac{1}{N_{\max}} \sum_{i=1}^{N_s} \log \Big( \sum_{j=1}^{M} \Pi_{j,i}\, \mathcal{N}(\Delta x_i, \Delta y_i \mid \mu_{x,j,i}, \mu_{y,j,i}, \sigma_{x,j,i}, \sigma_{y,j,i}, \rho_{xy,j,i}) \Big)$$

$$L_p = -\frac{1}{N_{\max}} \sum_{i=1}^{N_{\max}} \sum_{k=1}^{3} p_{k,i} \log(q_{k,i}), \ \ L_R = L_s + L_p. \tag{9}$$

Note that we discard the pdf parameters modelling the $(\Delta x, \Delta y)$ points beyond $N_s$ when calculating $L_s$, while $L_p$ is calculated using all of the pdf parameters modelling the $(p_1, p_2, p_3)$ points until $N_{\max}$. Both terms are normalized by the total sequence length $N_{\max}$. We found this methodology of loss calculation to be more robust and allows the model to easily learn when it should stop drawing, unlike the earlier mentioned method of assigning importance weightings to $p_1$, $p_2$, and $p_3$.

The Kullback-Leibler (KL) divergence loss term measures the difference between the distribution of our latent vector $z$, to that of an IID Gaussian vector with zero mean and unit variance. Optimizing for this loss term allows us to minimize this difference. We use the result in (Kingma & Welling, 2013), and calculate the KL loss term, $L_{KL}$, normalized by number of dimensions $N_z$ of $z$:

$$L_{KL} = -\frac{1}{2N_z}\Big(1 + \hat{\sigma} - \mu^2 - \exp(\hat{\sigma})\Big). \tag{10}$$

The loss function in Equation 11 is a weighted sum of both the $L_R$ and $L_{KL}$ loss terms:

$$Loss = L_R + w_{KL} L_{KL}. \tag{11}$$

There is a tradeoff between optimizing for one term over the other. As $w_{KL} \to 0$, our model approaches a pure autoencoder, sacrificing the ability to enforce a prior over our latent space while obtaining better reconstruction loss metrics. Note that for unconditional generation, where our model is the standalone decoder, there will be no $L_{KL}$ term as we only optimize for $L_R$.

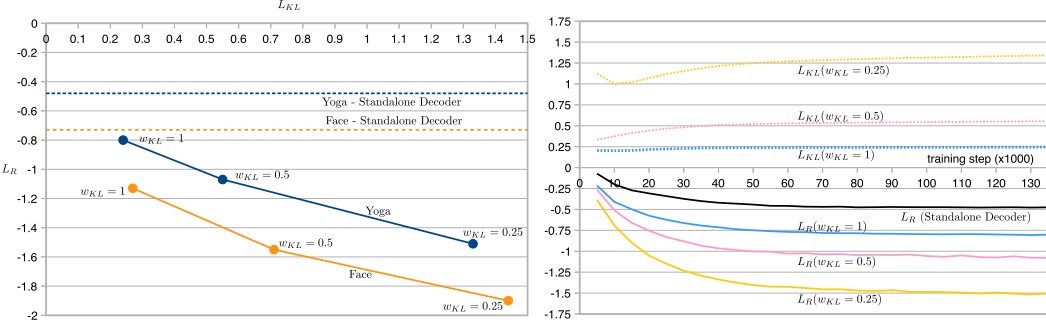

Figure 4: Tradeoff between $L_R$ and $L_{KL}$, for two models trained on single class datasets (left). Validation Loss Graph for models trained on the Yoga dataset using various $w_{KL}$ (right).

Figure 4 illustrates the tradeoff between different settings of $w_{KL}$ and the resulting $L_R$ and $L_{KL}$ metrics on the test set, along with the $L_R$ metric on a standalone decoder RNN for comparison. As the unconditional model does not receive any prior information about the entire sketch it needs to generate, the $L_R$ metric for the standalone decoder model serves as an upper bound for various conditional models using a latent vector.

## 4 EXPERIMENTS

We conduct several experiments with `sketch-rnn` for both conditional and unconditional vector image generation. We train `sketch-rnn` on various `QuickDraw` classes using various settings for $w_{KL}$ and record the breakdown of losses. To experiment with a diverse set of classes with varying complexities, we select the cat, pig, face, firetruck, garden, owl, mosquito and yoga class. We also experiment on multi-class datasets by concatenating different classes together to form (cat, pig) and (crab, face, pig, rabbit). The results for test set evaluation on various datasets are displayed in Table 1.

The `sketch-rnn` model treats the RNN cell as an abstract component. In our experiments, we use Long Short-Term Memory (LSTM) (Hochreiter & Schmidhuber, 1997) as the encoder RNN. For the decoder RNN, we use HyperLSTM, as this type of RNN cell excels at sequence generation tasks (Ha et al., 2017). The ability for HyperLSTM to spontaneously augment its own weights enables it to adapt to many different regimes in a large diverse dataset. Please see the Appendix for more details.

| Dataset | $w_{KL} = 1.00$ | | $w_{KL} = 0.50$ | | $w_{KL} = 0.25$ | | Decoder Only |
|---|---|---|---|---|---|---|---|
| | $L_R$ | $L_{KL}$ | $L_R$ | $L_{KL}$ | $L_R$ | $L_{KL}$ | $L_R$ |
| cat | -0.98 | 0.29 | -1.33 | 0.70 | -1.46 | 1.01 | -0.57 |
| pig | -1.14 | 0.22 | -1.37 | 0.49 | -1.52 | 0.80 | -0.82 |
| cat, pig | -1.02 | 0.22 | -1.24 | 0.49 | -1.50 | 0.98 | -0.75 |
| crab, face, pig, rabbit | -0.91 | 0.22 | -1.04 | 0.40 | -1.47 | 1.17 | -0.67 |
| face | -1.13 | 0.27 | -1.55 | 0.71 | -1.90 | 1.44 | -0.73 |
| firetruck | -1.24 | 0.22 | -1.26 | 0.24 | -1.78 | 1.10 | -0.90 |
| garden | -0.79 | 0.20 | -0.81 | 0.25 | -0.99 | 0.54 | -0.62 |
| owl | -0.93 | 0.20 | -1.03 | 0.34 | -1.29 | 0.77 | -0.66 |
| mosquito | -0.67 | 0.30 | -1.02 | 0.66 | -1.41 | 1.54 | -0.34 |
| yoga | -0.80 | 0.24 | -1.07 | 0.55 | -1.51 | 1.33 | -0.48 |

Table 1: Loss figures ($L_R$ and $L_{KL}$) for various $w_{KL}$ settings.

The relative loss numbers are consistent with our expectations. We see that the reconstruction loss term $L_R$ decreases as we relax the $w_{KL}$ parameter controlling the weight for the KL loss term, and meanwhile the KL loss term $L_R$ increases as a result. The $L_R$ for the conditional model is strictly less than the unconditional, standalone decoder model. In Figure 4 (right), we plot validation-set loss graphs for on the yoga class for models with various $w_{KL}$ settings. As $L_R$ decreases, the $L_{KL}$ term tends to increase due to the tradeoff between $L_R$ and $L_{KL}$.

### 4.1 CONDITIONAL RECONSTRUCTION

We qualitatively assess the reconstructed sketch $S'$ given an input sketch $S$. In Figure 5 (left), we sample several reconstructions at various levels of temperature $\tau$ using a model trained on the single cat class, starting at 0.01 on the left and linearly increasing to 1.0 on the right. The reconstructed cat sketches have similar properties as the input image, and occasionally add or remove details such as a whisker, a mouth, a nose, or the orientation of the tail.

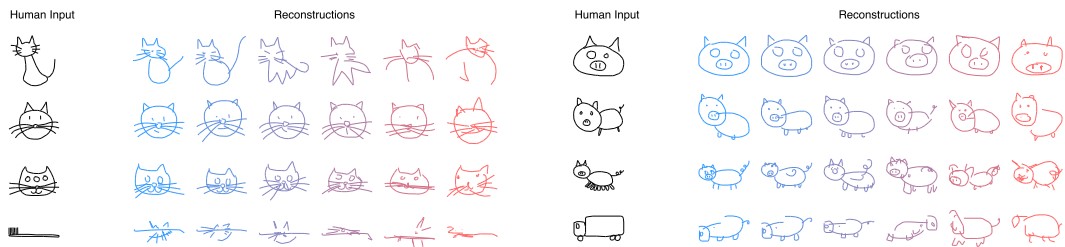

Figure 5: Conditional generation of cats (left) and pigs (right).

When presented with a non-standard image of a cat, such as a cat's face with three eyes, the reconstructed cat only has two eyes. If we input a sketch from another image class, such a toothbrush, the model seemingly generate sketches with similar orientation and properties as the toothbrush input image, but with some cat-like features such as cat ears, whiskers or feet. We perform a similar experiment with a model trained on the pig class, as shown in Figure 5 (right).

## 4.2 LATENT SPACE INTERPOLATION

By interpolating between latent vectors, we can visualize how one image morphs into another image by visualizing the reconstructions of the interpolations. As we enforce a Gaussian prior on the latent space, we expect fewer *gaps* in the space between two encoded latent vectors. We expect a model trained using a higher $w_{KL}$ setting to produce images that are closer to the data manifold given a spherically interpolated (White, 2016) latent vector $z$, compared to another model trained with a lower $w_{KL}$.

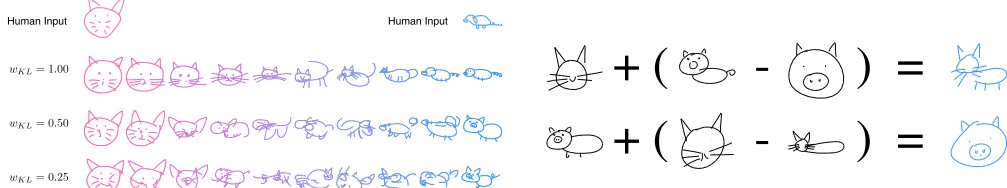

Figure 6: Latent space interpolation between cat and pig using with various $w_{KL}$ settings (left). Sketch Drawing Analogies (right).

To demonstrate this, we train several models using various $w_{KL}$, on a dataset consisting of both cat and pigs, and we encode two distinct images from the test set - a cat face and a full pig. Figure 6 (left) shows the reconstructed images from the interpolated latent vectors between the two original images. As expected, models trained with higher $w_{KL}$ produce more coherent interpolated images.

## 4.3 SKETCH DRAWING ANALOGIES

The interpolation example in Figure 6 (left) suggests that the latent vector $z$ encode conceptual features of a sketch. Can we use these features to augment other sketches without such features – for example, adding a body to a cat's head? Indeed, we find that sketch drawing analogies are possible for models trained with low $L_{KL}$ numbers. Given the smoothness of the latent space, where any interpolated vector between two latent vectors results in a coherent sketch, we can perform vector arithmetic on the latent vectors encoded from different sketches and explore how the model organizes the latent space to represent different concepts in the manifold of generated sketches.

For example, as shown in Figure 6 (right), we can subtract the latent vector of an encoded pig head from the latent vector of a full pig, to arrive at a vector that represents a body. Adding this difference to the latent vector of a cat head results in a full cat (i.e. cat head + body = full cat). We repeat the experiment to remove the body of a full pig. These drawing analogies allow us to explore how the model organizes its latent space to represent different concepts in the manifold of generated sketches.

## 4.4 PREDICTING DIFFERENT ENDINGS OF INCOMPLETE SKETCHES

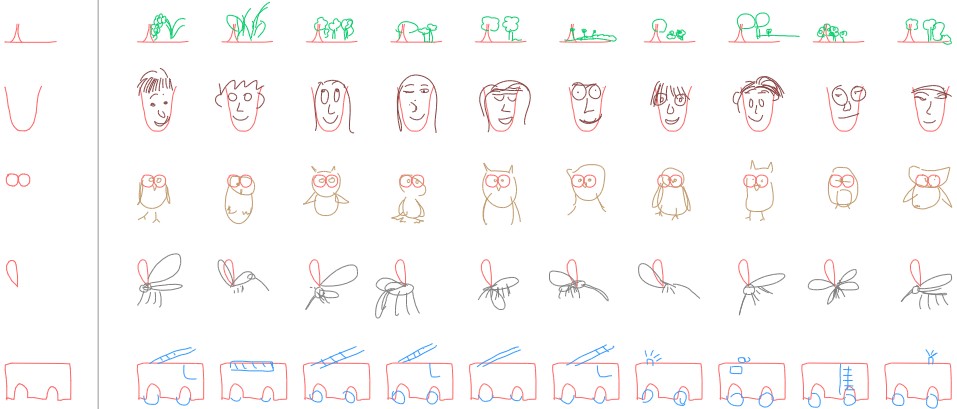

Figure 7: `sketch-rnn` predicting possible endings of various incomplete sketches (the red lines).

We can use `sketch-rnn` to finish an incomplete sketch. By using the decoder RNN as a standalone model, we can generate a sketch that is conditioned on the previous points. We use the decoder RNN to first *encode* an incomplete sketch into a hidden state $h$. Afterwards, we generate the remaining points of the sketch using $h$ as the initial hidden state. We show results in Figure 7 using decoder-only models trained on individual classes, and sample completions by setting $\tau = 0.8$.

## 5 APPLICATIONS AND FUTURE WORK

We believe `sketch-rnn` will enable many creative applications. Even the decoder-only model trained on various classes can assist the creative process of an artist by suggesting many possible ways of finishing a sketch, helping artists expand their imagination. In the conditional model, exploring the latent space between different objects can potentially enable artists to find interesting intersections and relationships between different drawings. Even in the simplest use, pattern designers can apply `sketch-rnn` to generate a large number of similar, but unique designs for textile or wallpaper prints.

As we saw earlier in Section 4.1, a model trained to draw pigs can be made to draw pig-like trucks if given an input sketch of a truck. We can extend this result to applications that might help creative designers come up with abstract designs that can resonate more with their target audience. For instance, in Figure 8 (right), we feed sketches of four different chairs into our cat-drawing model to produce four "chair-like cats". We can even interpolate between the four images to explore the latent space of chair-like cats, and select from a large grid of generated designs.

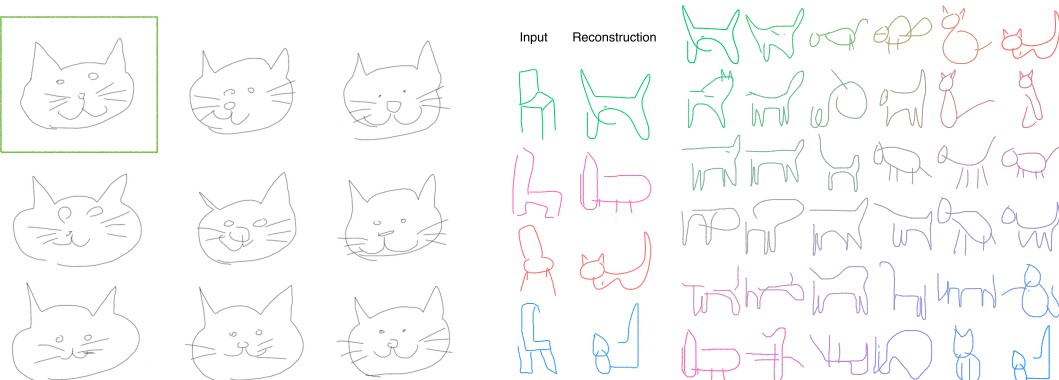

Figure 8: Generating similar, but unique sketches based on a single human sketch in the box (left). Latent space of generated cats conditioned on sketch drawings of chairs (right).

A model trained on higher quality sketches may find its way into educational applications that can help teach students how to draw. Even with the simple sketches in `QuickDraw`, the authors of this work have become much more proficient at drawing animals, insects, and various sea creatures after conducting these experiments. A related application is to encode a crude, poorly sketched drawing and generate more aesthetically looking reproductions by using a model trained with a high $w_{KL}$ setting and sampling with a low temperature $\tau$ to produce a more coherent version of the drawing. In the future, we can also investigate augmenting the latent vector in the direction that maximizes the aesthetics of the drawing by incorporating user-rating data into the training process.

Combining hybrid variations of sequence-generation models with unsupervised, cross-domain pixel image generation models, such as Image-to-Image models (Dong et al., 2017; Kim et al., 2017; Liu et al., 2017), is another exciting direction that we can explore. We can already combine this model with supervised, cross-domain models such as Pix2Pix (Isola et al., 2016), to occasionally generate photo realistic cat images from generated sketches of cats. The opposite direction of converting a photograph of a cat into an unrealistic, but similar looking sketch of a cat composed of a minimal number of lines seems to be a more interesting problem.

# 6 CONCLUSION

In this work, we develop a methodology to model sketch drawings using recurrent neural networks. `sketch-rnn` is able to generate possible ways to finish an existing, but unfinished sketch drawing. Our model can also encode existing sketches into a latent vector, and generate similar looking sketches conditioned on the latent space. We demonstrate what it means to interpolate between two different sketches by interpolating between its latent space, and also show that we can manipulate attributes of a sketch by augmenting the latent space. We demonstrate the importance of enforcing a prior distribution on the latent vector for coherent vector image generation during interpolation. By making available a large dataset of sketch drawings, we hope to encourage further research and development in the area of generative vector image modelling.

# 7 ACKNOWLEDGEMENTS

We thank Ian Johnson, Jonas Jongejan, Martin Wattenberg, Mike Schuster, Thomas Deselaers, Ben Poole, Kyle Kastner, Junyoung Chung and Kyle McDonald for their help with this project. This work was done as part of the Google Brain Residency program (`g.co/brainresidency`).

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

## A APPENDIX

### A.1 DATASET DETAILS

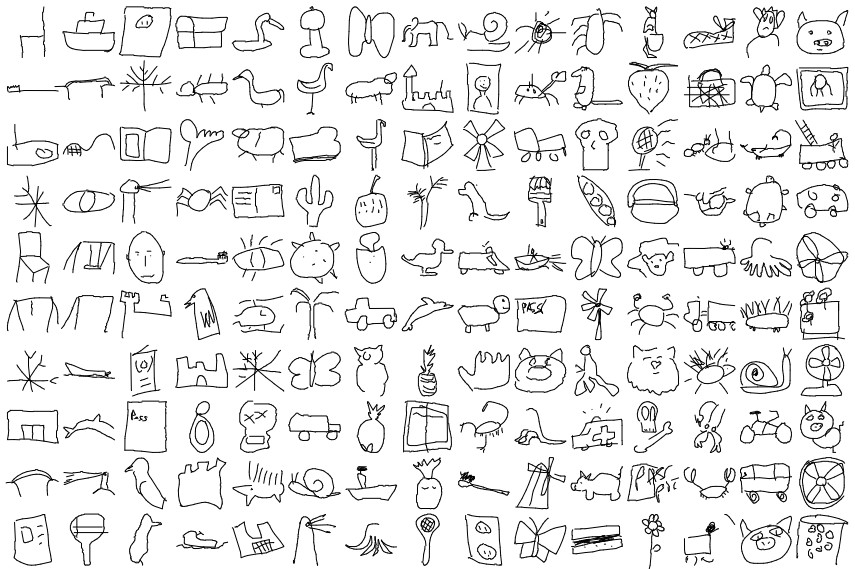

Figure 9: Example sketch drawings from `QuickDraw` dataset.

The data from `QuickDraw` (Jongejan et al., 2016) expands daily, and every so often new classes are added to the game. As such, the `QuickDraw` dataset now consists of hundreds of classes, from 75 classes initially, in Table 2. In total, there are $\sim 50$ million sketches in the released dataset, although for the purpose of constructing an organized dataset for research purposes, we have limited the number of sketches in each class.

| alarm clock | ambulance | angel | ant | barn |
|---|---|---|---|---|
| basket | bee | bicycle | book | bridge |
| bulldozer | bus | butterfly | cactus | castle |
| cat | chair | couch | crab | cruise ship |
| dolphin | duck | elephant | eye | face |
| fan | fire hydrant | firetruck | flamingo | flower |
| garden | hand | hedgehog | helicopter | kangaroo |
| key | lighthouse | lion | map | mermaid |
| octopus | owl | paintbrush | palm tree | parrot |
| passport | peas | penguin | pig | pineapple |
| postcard | power outlet | rabbit | radio | rain |
| rhinoceros | roller coaster | sandwich | scorpion | sea turtle |
| sheep | skull | snail | snowflake | speedboat |
| spider | strawberry | swan | swing set | tennis racquet |
| the mona lisa | toothbrush | truck | whale | windmill |

Table 2: Initial 75 `QuickDraw` classes used for this work.

Each class consists of 70K training samples and 2.5K validation and test samples. Stroke simplification using the Ramer–Douglas–Peucker algorithm (Douglas & Peucker, 1973) with a parameter of $\epsilon = 2.0$ has been applied to simplify the lines. The data was originally recorded in pixel-dimensions, so we normalized the offsets $(\Delta x, \Delta y)$ using a single scaling factor. This scaling factor was calculated to adjust the offsets in the training set to have a standard deviation of 1. For simplicity, we do not normalize the offsets $(\Delta x, \Delta y)$ to have zero mean, since the means are already relatively small. Figure 10 shows a training example before normalization of $(\Delta x, \Delta y)$ data columns.

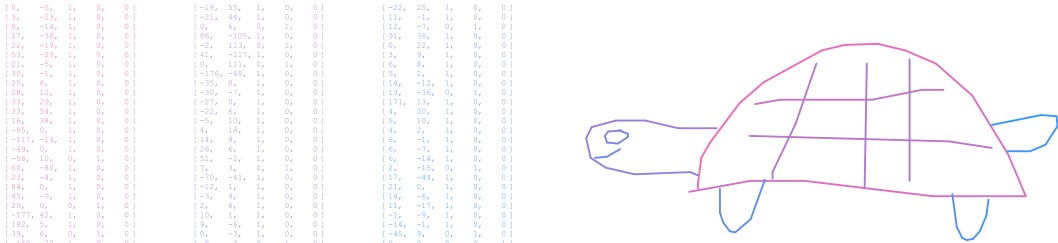

Figure 10: A sample sketch, as a sequence of $(\Delta x, \Delta y, p_1, p_2, p_3)$ points and in rendered form. In the rendered sketch, the line color corresponds to the sequential stroke ordering.

## A.2 TRAINING DETAILS

As a recap from the main text, we defined the Reconstruction loss term $L_R$ as:

$$L_s = -\frac{1}{N_{\max}} \sum_{i=1}^{N_s} \log \Big( \sum_{j=1}^{M} \Pi_{j,i} \, \mathcal{N}(\Delta x_i, \Delta y_i \mid \mu_{x,j,i}, \mu_{y,j,i}, \sigma_{x,j,i}, \sigma_{y,j,i}, \rho_{xy,j,i}) \Big)$$

$$L_p = -\frac{1}{N_{\max}} \sum_{i=1}^{N_{\max}} \sum_{k=1}^{3} p_{k,i} \log(q_{k,i})$$

$$L_R = L_s + L_p. \tag{12}$$

We also defined the KL loss term $L_{KL}$ as:

$$L_{KL} = -\frac{1}{2N_z}\Big(1 + \hat{\sigma} - \mu^2 - \exp(\hat{\sigma})\Big). \tag{13}$$

The loss function in Equation 14 is a weighted sum of both the $L_R$ and $L_{KL}$ loss terms:

$$Loss = L_R + w_{KL} L_{KL}. \tag{14}$$

While the loss function in Equation 14 can be used during training, we find that annealing the KL term in the loss function (Equation 15) produced better results. This modification is only used for model training, and the original loss function in Equation 14 is still used to evaluate validation and test sets, and for early stopping.

$$\eta_{step} = 1 - (1 - \eta_{min})R^{step}$$

$$Loss_{train} = L_R + w_{KL}\eta_{step}\max(L_{KL}, KL_{min}) \tag{15}$$

We find that annealing the KL loss term generally results in better losses. Annealing the $L_{KL}$ term in the loss function directs the optimizer to first focus more on the reconstruction term in Equation 12, which is the more difficult loss term of the model to optimize for, before having to deal with optimizing for the KL loss term in Equation 13, a far simpler expression in comparison. This approach has been used in (Bowman et al., 2015; Kaae Sønderby et al., 2016; Kingma et al., 2016). Our annealing term $\eta_{step}$ starts at $\eta_{min}$ (typically 0 or 0.01) at training step 0, and converges to 1 for large training steps. $R$ is a term close to, but less than 1.

If the distribution of $z$ is close enough to $\mathcal{N}(0, I)$, we can sample sketches from the decoder using randomly sampled $z$ from $\mathcal{N}(0, I)$ as the input. In practice, we find that going from a larger $L_{KL}$ value ($L_{KL} > 1.0$) to a smaller $L_{KL}$ value of 0.3 generally results in a substantial increase in the quality of sampled images using randomly sampled $z \sim \mathcal{N}(0, I)$. However, going from $L_{KL} = 0.3$ to $L_{KL}$ values closer to zero does not lead to any further noticeable improvements. Hence we find it useful to put a floor on $L_{KL}$ in the loss function by enforcing $\max(L_{KL}, KL_{min})$ in Equation 15.

The $KL_{min}$ term inside the max operator is typically set to a small value such as 0.10 to 0.50. This term will encourage the optimizer to put less focus on optimizing for the KL loss term $L_{KL}$ once it is low enough, so we can obtain better metrics for the reconstruction loss term $L_R$. This approach is similar to the approach described in (Kingma et al., 2016) as free bits, where they apply the max operator separately inside each dimension of the latent vector $z$.

## A.3 MODEL CONFIGURATION

Our encoder and decoder RNNs consist of 512 and 2048 nodes respectively. In our model, we use $M = 20$ mixture components for the decoder RNN. The latent vector $z$ has $N_z = 128$ dimensions. We apply Layer Normalization (Ba et al., 2016) to our model, and during training apply recurrent dropout [9] with a keep probability of 90%. We train the model with batch sizes of 100 samples, using Adam (Kingma & Ba, 2015) with a learning rate of 0.0001 and gradient clipping of 1.0. All models are trained with $KL_{min} = 0.20, R = 0.99999$. During training, we perform simple data augmentation by multiplying the offset columns $(\Delta x, \Delta y)$ by two IID random factors chosen uniformly between 0.90 and 1.10. Unless mentioned otherwise, all experiments are conducted with $w_{KL} = 1.00$.

## A.4 MODEL LIMITATIONS

Although sketch-rnn can model a large variety of sketch drawings, there are several limitations in the current approach we wish to highlight. For most single-class datasets, sketch-rnn is capable of modelling sketches up to around 300 data points. The model becomes increasingly difficult to train beyond this length. For our dataset, we applied the Ramer–Douglas–Peucker algorithm (Douglas & Peucker, 1973) to simplify the strokes of the sketch data to less than 200 data points while still keeping most of the important visual information of each sketch.

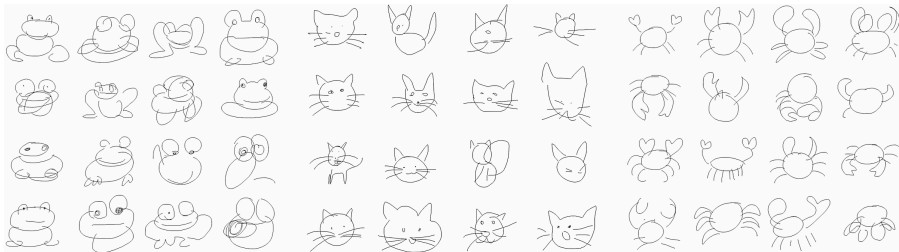

Figure 11: Unconditional generated sketches of frogs, cats, and crabs at $\tau = 0.8$

For more complicated classes of images, such as mermaids or lobsters, the reconstruction loss metrics are not as good compared to simpler classes such as ants, faces or firetrucks. The models trained on these more challenging image classes tend to draw smoother, more circular line segments that do not resemble individual sketches, but rather resemble an averaging of many sketches in the training set. We can see some of this artifact in the frog class, in Figure 11. This smoothness may be analogous to the blurriness effect produced by a Variational Autoencoder (Kingma & Welling, 2013) that is trained on pixel images. Depending on the use case of the model, smooth circular lines can be viewed as aesthetically pleasing and a desirable property.

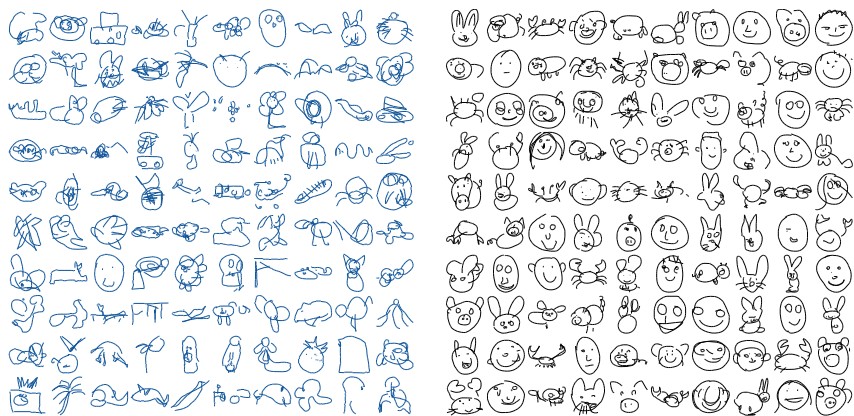

Figure 12: Unconditional generations from model trained on 75 classes (left),
and from model trained on crab, face, pig and rabbit classes (right).

While both conditional and unconditional models are capable of training on datasets consisting of several classes, such as (cat, pig), and (crab, face, pig, rabbit), `sketch-rnn` is ineffective at modelling a large number of classes simultaneously. In Figure 12, we sample sketches using an unconditional model trained on 75 classes, and a model trained on 4 classes. The samples generated from the 75-class model are incoherent, with individual sketches displaying features from multiple classes. The four-class unconditional model usually generates samples of a single class, but occasionally also combines features from multiple classes. In the future, we will explore incorporating class information outside of the latent space to handle the modelling of a large number of classes simultaneously.

## A.5 MULTI-SKETCH DRAWING INTERPOLATION

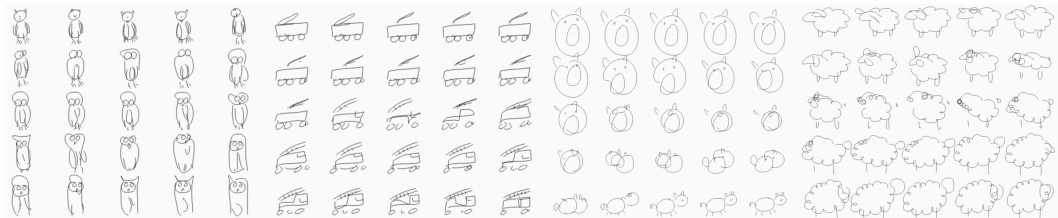

Figure 13: Example of conditional generated sketches with single class models.
Latent space interpolation from left to right, and then top to bottom.

In addition to interpolating between two sketches, like in Figure13, we can also visualize the interpolation between four sketches in latent space to gain further insight from the model. In this section we show more examples conditionally generated with `sketch-rnn`. We take four generated images, place them on four corners of a grid, and populate the rest of the grid using the interpolation of the latent vectors at the corners. Figure 14 shows two examples of this four-way interpolation, using models trained on both (cat, pig) classes, and face class. All samples generated with $\tau = 0.1$.

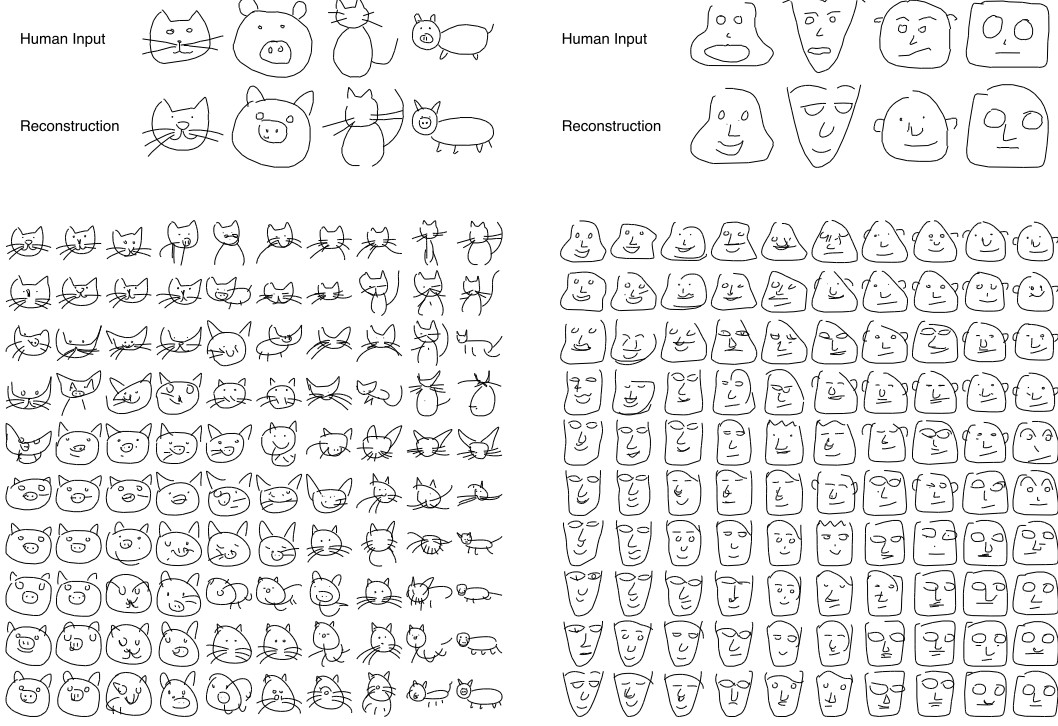

Figure 14: Example input sketches and `sketch-rnn` generated reproductions (Top).
Latent space interpolation between the four reproduced sketches (Bottom).

The left most figure of Figure 15 visualizes the interpolation between a full pig, a rabbit's head, a crab, and a face, using a model trained on these four classes. In certain parts of the space between a crab and a face is a rabbit's head, and we see that the ears of the rabbit becomes the crab's claws. Applying the model on the yoga class, it is interesting to see how one yoga position slowly transitions to another via a set of interpolated yoga positions generated by the model. For visual effect, we also interpolate between four distinct colors, and color each sketch using a unique interpolated color.

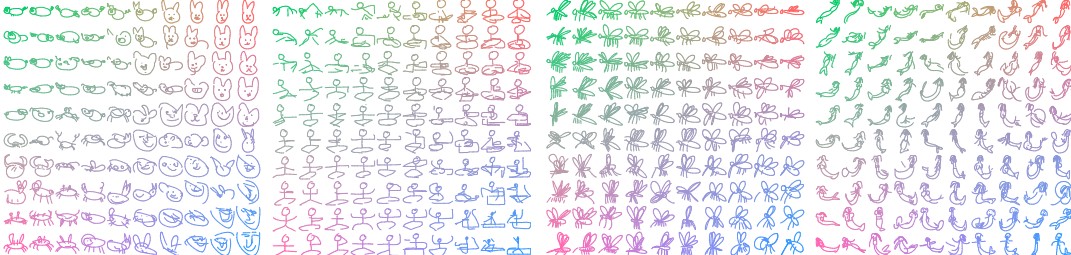

Figure 15: Interpolation of (pig, rabbit, crab and face), yoga poses, mosquitoes and mermaids. We also interpolate between four distinct colors for visual effect.

We also construct latent space interpolation examples for the mosquito class and the mermaid class, in the last two grids Figure 15. We see that the model can interpolate between concepts such as style of wings, leg counts, and orientation. In Figure 16 below, we show more interpolation examples of other classes from the dataset.

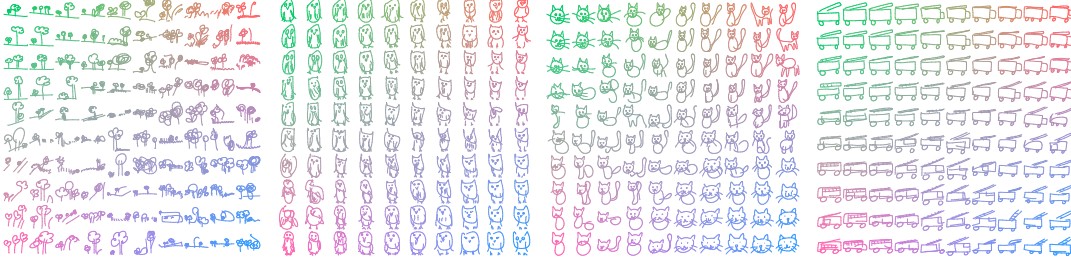

Figure 16: Latent space interpolation between four generated gardens, owls, cats, and firetrucks.

## A.6 WHICH LOSS CONTROLS IMAGE COHERENCY?

We would like to question the relative importance of the reconstruction loss term $L_R$, relative to the KL loss term $L_{KL}$, when our goal is to produce higher quality image reconstructions. While our reconstruction loss term $L_R$ optimizes for the log-likelihood of the set of strokes that make up a sketch, this metric alone does not give us any guarantee that a model with a lower $L_R$ number will produce higher quality reconstructions compared to a model with a higher $L_R$ number.

For example, imagine a simple sketch of an face, ☺, where most of the data points of $S$ are be used to represent the head, and only a minority of points represent facial features such as the eyes and mouth. It is possible to reconstruct the face with incoherent facial features, and yet still score a lower $L_R$ number compared to another reconstruction with a coherent and similar face, if the edges around the incoherent face are generated more precisely.

In Figure 17, we compare the reconstructed images generated using models trained with various $w_{KL}$ settings. In the first three examples from the left, we train our model on a dataset consisting of four image classes (crab, face, pig, rabbit). We deliberately sketch input drawings that contain features of two classes, such as a rabbit with a pig mouth and pig tail, a person with animal ears, and a rabbit with crab claws. We see that the model trained using higher $w_{KL}$ weights, tend to generate sketches with features of a single class that look more coherent, despite having lower $L_{KL}$ numbers. For instance, the model with $w_{KL} = 1.00$ omit pig features, animal ears, and crab claws from its reconstructions. In contrast, the model with $w_{KL} = 0.25$, with higher $L_{KL}$, but lower $L_R$ numbers tries to keep both inconsistent features, while generating sketches that look less coherent.

In the last three examples in Figure 17, we repeat the experiment on models trained on single-class images, and see similar results even when we deliberately choose input samples from the test set with noisier lines.

If we look at the interpolations produced in the latent space interpolation examples from Section 4.2 in the main text, models with better KL loss terms also generate more meaningful reconstructions from the interpolated space between two latent vectors. This suggests the latent vector for models with lower $L_{KL}$ control more meaningful parts of the drawings, such as controlling whether the sketch is an animal head only or a full animal with a body, or whether to draw a cat head or a pig head. Altering such latent vectors can allow us to directly manipulate these animal features. Conversely, altering the latent codes of models with higher $L_{KL}$ results in scattered movement of individual line segments, rather than alterations of meaningful conceptual features of the animal.

This result is consistent with incoherent reconstructions seen in Figure 17. With a lower $L_{KL}$, the model is likely to generate coherent images given any random $z$. Even with a non-standard, or noisy, input image, the model will still encode a $z$ that produces coherent images. For models with lower $L_{KL}$ numbers, the encoded latent vectors contain conceptual features belonging to the input image, while for models with higher $L_{KL}$ numbers, the latent vectors merely encode information about specific line segments. This observation suggests that when using sketch-rnn on a new dataset, we should first try different $w_{KL}$ settings to evaluate the tradeoff between $L_R$ and $L_{KL}$, and then choose a setting for $w_{KL}$ (and $KL_{min}$) that best suit our requirements.

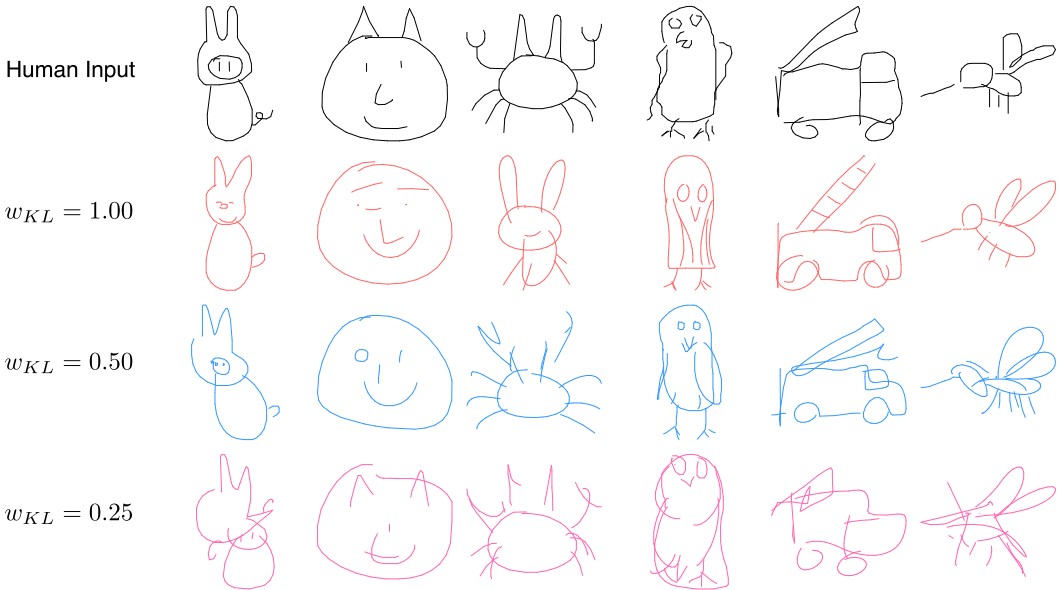

Figure 17: Reconstructions of sketch drawings using models with various $w_{KL}$ settings.

