# OpenReview forum: "A Neural Representation of Sketch Drawings"
_ICLR.cc/2018/Conference — Accept (Poster)_

### Official Review · AnonReviewer2 · 2017-11-27
**Exciting application that employs deep learning tricks to get exciting results**

**Rating:** 8
**Confidence:** 4

**Review:**

This paper introduces a neural network architecture for generating sketch drawings. The authors propose that this is particularly interesting over generating pixel data as it emphasises more human concepts. I agree. The contribution of this paper of this paper is two-fold. Firstly, the paper introduces a large sketch dataset that future papers can rely on. Secondly, the paper introduces the model for generating sketch drawings.

The model is inspired by the variational autoencoder. However, the proposed method departs from the theory that justifies the variational autoencoder. I believe the following things would be interesting points to discuss / follow up:
- The paper preliminarily investigates the influence of the KL regularisation term on a validation data likelihood. It seems to have a negative impact for the range of values that are discussed. However, I would expect there to be an optimum. Does the KL term help prevent overfitting at some stage? Answering this question may help understand what influence variational inference has on this model.
- The decoder model has randomness injected in it at every stage of the RNN. Because of this, the latent state actually encodes a distribution over drawings, rather than a single drawing. It seems plausible that this is one of the reasons that the model cannot obtain a high likelihood with a high KL regularisation term. Would it help to rephrase the model to make the mapping from latent representation to drawing more deterministic? This definitely would bring it closer to the way the VAE was originally introduced.
- The unconditional generative model *only* relies on the "injected randomness" for generating drawings, as the initial state is initialised to 0. This also is not in the spirit of the original VAE, where unconditional generation involves sampling from the prior over the latent space.

I believe the design choices made by the authors to be valid in order to get things to work. But it would be interesting to see why a more straightforward application of theory perhaps *doesn't* work as well (or whether it works better). This would help interesting applications inform what is wrong with current theoretical views.

Overall, I would argue that this paper is a clear accept.

---

> ### Author Response · Authors · 2017-12-24
> **Thank you for the insightful comments.**
>
> It is true that our design choices were made to get things to work, and despite this, the current model still has many issues that can be improved upon in the future. For example, the model does not perform well for long sequence length. We needed to use the Ramer–Douglas–Peucker (RDP) algorithm to simplify the strokes, which also made the data more consistent for the RNN. We have included these details and tried to put information about model limitations in the A1 Dataset Details section.
>
> With your feedback, and also along with the feedback from AnonReviewer3, we have added a short section in A6 that examines the tradeoff between likelihood and KL. We examine what happens qualitatively to the sketches as we vary the weighting on the KL term. Hopefully this will be a good starting point for future work.
>
> In future work we will explore in depth the regularisation methodology - perhaps KL is not the best one to use and we wish to explore alternative approaches, for example alternatives outlined in [1].
>
> [1] InfoVAE: Information Maximizing Variational Autoencoders (https://arxiv.org/abs/1706.02262).

---

### Official Review · AnonReviewer3 · 2017-11-28
**Interesting problem and good approach to solve it.**

**Rating:** 8
**Confidence:** 4

**Review:**

The paper aims tackles the problem of generate vectorized sketch drawings by using a RNN-variational autoencoder. Each node is represented with (dx, dy) along with one-hot representation of three different drawing status. A bi-directional LSTM is used to encode latent space in the training stage. Auto-regressive VAE is used for decoding.

Similar to standard VAEs, log-likelihood has bee used as the data-term and the KL divergence between latent space and Gaussian prior is the regularisation term.

Pros:
- Good solution to an interesting problem.
- Very interesting dataset to be released.
- Intensive experiments to validate the performance.

Cons:
- I am wondering whether the dataset contains biases regarding (dx, dy). In the data collection stage, how were the points lists generated from pen strokes?  Did each points are sampled from same travelling distance or according to the same time interval?  Are there any other potential biases brought because the data collection tools?
- Is log-likelihood a good loss here? Think about the case where the sketch is exactly the same but just more points are densely sampled along the pen stroke. How do you deal with this case?
- Does the dataset contain more meta-info that could be used for other tasks beyond generation, e.g. segmentation, classification, identification, etc.?

---

> ### Author Response · Authors · 2017-12-24
> **Thanks for your comments and feedback.**
>
> Regarding the data collection, we have used the Ramer–Douglas–Peucker (RDP) algorithm as a pre-processing step to simplify the strokes in the dataset. Using RDP, line strokes drawn very slowly (with many points) and drawn very swiftly with look similar after the simplification process. For example, if the user holds his or her finger on the screen in one location for many seconds while sketching something, many points will be generated at a single location, but the simplification method will collapse those points as a single point. We put details of the data collection and stroke simplification in A1. Dataset Details.
>
> The dataset will contain meta-info, such as country information, timestamp, and class, so we hope it can be used for classification experiments, and even for exploring cultural biases in the way we draw.
>
> You raise an interesting point about whether log-likelihood is a good loss, especially in the case "where the sketch is exactly the same but just more points are densely sampled along the pen stroke". Based on your feedback, and also the feedback of AnonReviewer2, we have added a section in A6 "Which Loss Controls Image Coherency?", where we look at whether the KL loss term helps in such cases.
>
> We explore the tradeoff between varying weights of the KL loss term and see that increasing the KL weighting produces qualitatively better reconstructions, despite having a lower log-likelihood loss number. We will investigate alternative loss formulations in future work, perhaps looking at adversarial methods, but we hope this will be a good start in that direction.

---

### Official Review · AnonReviewer1 · 2017-11-29

**Rating:** 5
**Confidence:** 4

**Review:**

The paper presents both a novel large dataset of sketches and a new rnn architecture to generate new sketches.

+ new and large dataset
+ novel algorithm
+ well written
- no evaluation of dataset
- virtually no evaluation of algorithm
- no baselines or comparison

The paper is well written, and easy to follow. The presented algorithm sketch-rnn seems novel and significantly different from prior work.
In addition, the authors collected the largest sketch dataset, I know of. This is exciting as it could significantly push the state of the art in sketch understanding and generation.

Unfortunately the evaluation falls short. If the authors were to push for their novel algorithm, I'd have expected them to compare to prior state of the art on standard metrics, ablate their algorithm to show that each component is needed, and show where their algorithm shines and where it falls short.
For ablation, the bare minimum includes: removing the forward and/or reverse encoder and seeing performance drop. Remove the variational component, and phrasing it simply as an auto-encoder. Table 1 is good, but not sufficient. Training loss alone likely does not capture the quality of a sketch.
A comparison the Graves 2013 is absolutely required, more comparisons are desired.
Finally, it would be nice to see where the algorithm falls short, and where there is room for improvement.

If the authors wish to push their dataset, it would help to first evaluate the quality of the dataset. For example, how well do humans classify these sketches? How diverse are the sketches? Are there any obvious modes? Does the discretization into strokes matter?
Additionally, the authors should present a few standard evaluation metrics they would like to compare algorithms on? Are there any good automated metrics, and how well do they correspond to human judgement?

In summary, I'm both excited about the dataset and new architecture, but at the same time the authors missed a huge opportunity by not establishing proper baselines, evaluating their algorithm, and pushing for a standardized evaluation protocol for their dataset. I recommend the authors to decide if they want to present a new algorithm, or a new dataset and focus on a proper evaluation.

---

> ### Author Response · Authors · 2018-01-03
> **Thanks for your feedback.**
>
> We agree with the reviewer that we try to do many things in this paper - introduce a method for generating vector images and introducing a large dataset of vector drawings, and entered a less-explored area with few established evaluation metrics.
>
> As the dataset, and area of vector image modelling is new, our architecture was designed with simplicity in mind to become a baseline for future work. You mentioned Graves 2013, a work that we actually based our method on. Specifically: once we take away the encoder, and generate images unconditionally using the decoder model, it is identical to the autoregressive modelling approach taken in Graves 2013. The only minor difference is we needed to model the "end of drawing" probability and have appended the model to output that as well. You are right to mention that we should compare our encoder to a forward-only RNN, to see the metrics drop, although in practice we would argue that most practitioners would choose the bi-directional method from the onset especially when the length of the data becomes longer, and the architecture and task makes it possible to use a non-causal model.  For example, while Graves 2013 uses a unidirectional LSTM for decoder-only handwriting generation, Graves 2007 [1] uses a bidirectional LSTM for handwriting classification without comparing to a unidirectional one.
>
> You make some great points about the metrics relating to the dataset.  We are particularly interested in how the diversity and multi-modality of these drawings relate to issues like novelty and interpretability. In our view, these human perception and generation issues are complex and important enough to warrant their own future paper(s). To make it more convenient for future work on this dataset, we have also standardized the format and limited each class to have exactly 70K samples, 2.5K validation and test samples, rather than using the full extent of the dataset. We hope this standardisation will encourage future experiments in not just generation, but also classification, and also examination into cultural biases, diversity, modes, and human performance.
>
> [1] A. Graves, S. Fernández, M. Liwicki, H. Bunke and J. Schmidhuber. Unconstrained online handwriting recognition with recurrent neural networks. NIPS 2007, Vancouver, Canada.
>
> https://papers.nips.cc/paper/3213-unconstrained-on-line-handwriting-recognition-with-recurrent-neural-networks

---

### Decision · Program_Chairs · 2018-01-29
**ICLR 2018 Conference Acceptance Decision**

**Decision:**

Accept (Poster)

**Comment:**

This work presents a RNN tailored to generate sketch drawings. The model has novel elements and advances specific to the considered task, and allows for free generation as well as generation with (partial) input. The results are very satisfactory. Importantly, as part of this work a large dataset of sketch drawings is released. The only negative aspect is the insufficient evaluation, as pointed out by R1 who points out the need for baselines and evaluation metrics. R1’s concerns have been acknowledged by the authors but not really addressed in the revision. Still, this is a very interesting contribution.